# The Role of Gut–Liver Axis in Gut Microbiome Dysbiosis Associated NAFLD and NAFLD-HCC

**DOI:** 10.3390/biomedicines10030524

**Published:** 2022-02-23

**Authors:** Qian Song, Xiang Zhang

**Affiliations:** 1Department of Medicine and Therapeutics, Institute of Digestive Disease, The Chinese University of Hong Kong, Hong Kong 999077, China; songqian@link.cuhk.edu.hk; 2State Key Laboratory of Digestive Disease, Li Ka Shing Institute of Health Sciences, The Chinese University of Hong Kong, Shatin, Hong Kong 999077, China

**Keywords:** gut–liver axis, gut microbiome, metabolites, intestinal barrier, non-alcoholic fatty liver disease

## Abstract

Nonalcoholic fatty liver disease (NAFLD) is considered as one of the most prevalent chronic liver diseases worldwide due to the rapidly rising prevalence of obesity and metabolic syndrome. As a hepatic manifestation of metabolic disease, NAFLD begins with hepatic fat accumulation and progresses to hepatic inflammation, termed as non-alcoholic steatohepatitis (NASH), hepatic fibrosis/cirrhosis, and finally leading to NAFLD-related hepatocellular carcinoma (NAFLD-HCC). Accumulating evidence showed that the gut microbiome plays a vital role in the initiation and progression of NAFLD through the gut–liver axis. The gut–liver axis is the mutual communication between gut and liver comprising the portal circulation, bile duct, and systematic circulation. The gut microbiome dysbiosis contributes to NAFLD development by dysregulating the gut–liver axis, leading to increased intestinal permeability and unrestrained transfer of microbial metabolites into the liver. In this review, we systematically summarized the up-to-date information of gut microbiome dysbiosis and metabolomic changes along the stages of steatosis, NASH, fibrosis, and NAFLD-HCC. The components and functions of the gut–liver axis and its association with NAFLD were then discussed. In addition, we highlighted current knowledge of gut microbiome-based treatment strategies targeting the gut–liver axis for preventing NAFLD and its associated HCC.

## 1. Introduction

Non-alcoholic fatty liver disease (NAFLD) is characterized by lipid accumulation in more than 5% of hepatocytes [1]. It is a disease continuum from non-alcoholic fatty liver (simple steatosis) to non-alcoholic steatohepatitis (NASH), and finally to NAFLD-related hepatocellular carcinoma (NAFLD-HCC) [2]. NAFLD has become the most common cause of chronic liver disease worldwide, affecting 15–40% of general population [3]. In obese individuals, the prevalence of NAFLD can even reach to 90% [4]. NAFLD is considered as the hepatic component of metabolic syndrome and is closely related to metabolic diseases such as obesity, type 2 diabetes mellitus (T2DM), and atherosclerosis [5]. Based on this property, an international expert consensus statement suggested that NAFLD can be redefined to metabolic (dysfunction)-associated fatty liver disease (MAFLD) [6]. The diagnostic criteria of MAFLD include hepatic steatosis in addition to obesity, T2DM, or metabolic dysfunction [7]. However, there is a debate that the criteria of MAFLD are poorly applicable in real clinical practice as some non-obese, non-diabetic patients with hepatic steatosis could not be diagnosed due to the lack of laboratory tests for metabolic dysfunction [8].

Environmental and nutritional factors can usually account for the pathogenesis of NAFLD, both of which can contribute to the onset of NAFLD and its progression to NASH and NAFLD-HCC. Among environmental factors, gut microbiome dysbiosis is emerging as a crucial factor in the development of NAFLD. The gut microbiota is considered as an indispensable organ, which interacts with host cells for metabolism [9]. The balance of the gut microbiota community is essential to maintain the homeostasis of body metabolism. In 2004, a study demonstrated that the gut microbiota can regulate energy harvesting and energy storage from the diet [10]. This was an early study investigating the engagement of the gut microbiota in host metabolism regulation. Since then, emerging evidence has shown the critical effects of gut microbiota on the maintenance of the host metabolism [11]. Thus, the dysbiosis of the microbiota community is able to, directly and indirectly, influence the host metabolism [11]. Until now, aberrant intestinal microbiota have been reported to be associated with many metabolic disorders, including NAFLD [12].

The gut–liver axis is the mutual communication between the intestine and the liver (Figure 1). This axis is connected by portal circulation, the bile tract, as well as systematic circulation [13]. The liver obtains more than two-thirds of its blood from the gastrointestinal tract by the portal system. Through portal vein, intestine-derived bacteria and their components can easily reach the liver. In the liver, bacteria can stimulate hepatic immune cells, activate inflammation pathways, and eventually proceed to NAFLD/NAFLD-HCC [14]. This further confirms the crucial role of the gut–liver axis in the pathogenesis of NAFLD and NAFLD-HCC. Thus, the gut–liver axis can provide a therapeutic target for NAFLD. On one hand, improving the gut–liver axis can protect the liver from the pathogenic components in the intestine. On the other hand, probiotics and some beneficial microbial compositions can protect the liver via the gut–liver axis. Taken together, the gut–liver axis plays a critical role on mediating the function of gut microbiota in the progression of NAFLD liver (Figure 2). Elucidating the association of gut microbiota and gut–liver axis with NAFLD development can supply possible strategies for the prevention and treatment of NAFLD/NAFLD-HCC.

In the present review, we summarize the evidence of gut microbiota dysbiosis and microbial metabolites changes in NAFLD and NAFLD-HCC. Based on this, the components of the gut–liver axis and their association with NAFLD are illustrated in this review. In addition, the microbe-based treatment on NAFLD via targeting the gut–liver axis is provided.

## 2. Gut Microbiome Dysbiosis in NAFLD and NAFLD-HCC

### 2.1. Gut Microbiome Dysbiosis in Hepatic Steatosis

The gut microbiota is a crucial factor in the development of hepatic steatosis. For example, fecal transplantation experiment (FMT) from obese mice with hepatic steatosis to germ-free mice can induce NAFLD alterations, such as the elevated level of hepatic triglyceride and upregulation of genes related to lipogenesis and lipid uptake [15]. In addition, another study found that wild type mice can easily develop hepatic steatosis by co-housing mice with NASH. This phenotype has found to be associated with inflammasome-mediated gut dysbiosis [16]. Notably, FMT from obese patients with liver steatosis to mice can increase liver triglyceride accumulation within two weeks [17]. More interestingly, investigators found that obese infant mice with a western diet have excess weight gain and accelerate the progression of NAFLD [18]. This indicates that gut dysbiosis in maternal obesity-associated infants is critical to childhood NAFLD. On the basis of these studies, it is clear that the gut microbiota plays an indispensable role in contributing to the hepatic steatosis.

### 2.2. Gut Microbiome Dysbiosis in NASH/Fibrosis

Considering that the gut microbiota is involved in the pathophysiology of NAFLD development, microbiota dysbiosis can serve as a reliable non-invasive tool for the early diagnosis of NAFLD. In Europe, compared with healthy subjects, NAFLD patients have the high abundance of *Bradyrhizobium, Anaerococcus, Peptoniphilus, Propionibacterium acnes, Dorea*, and *Ruminococcus,* with the low abundance of *Oscillospira* and *Rikenellaceae* [19]. More interestingly, the microbiota dysbiosis types of NAFLD patients depend on various areas and sex. In a Chinese cohort, the genera *Lactobacillus, Oscillibacter*, and *Ruminiclostridium* have been found to be decreased in obese NAFLD patients, while *Faecalibacterium prausnitzii* was the only species that presents a different abundance between those with and without NAFLD [20]. For the women cohort, the abundance of several different genera such as *Subdoligranulum, Coprococcus,* and *Coprobacter* were negatively correlated with hepatic steatosis [17]. This provides evidence of gender differences in the gut microbiota among NAFLD patients, whereby the gut microbiota can be a biomarker in the context of gender backgrounds.

As a more aggressive type of NAFLD, NASH patients have a distinct gut microbiome compared with those who only have hepatic steatosis [18]. A cross-sectional study demonstrated that NASH patients had the higher abundance of fecal *Clostridium coccoides* and a lower percentage of *Bacteroidetes* compared to those with steatosis but without cancer [21]. Considering hepatic fibrosis is a severe stage of NASH that need clinical intervention, several studies compared the gut microbiome between non-fibrosis and fibrosis NAFLD patients. In these studies, investigators found an increased level of Bacteroides and a decreased level of several different genera, such as *Prevotella* [22,23]. Furthermore, Loomba et al. developed a universal gut microbiome-derived signature to predict NAFLD-cirrhosis. This study combined microbial species, age, and serum measures to make a comprehensive diagnostic signature for NAFLD-cirrhosis patients [24]. However, some studies have shown some controversial gut microbiome in NASH patients. For example, Boursier et al. found that *Prevotella* was decreased in F0/1 fibrosis stage NASH patients compared to F ≥ 2 fibrosis NASH patients, whereas Rau et al. demonstrated that patients with advanced fibrosis had a higher abundance of *Prevotella* [22,25]. This may be caused by the various genetic backgrounds of patients. Boursier et al. recruited French, whereas Rau et al. recruited Germans. Except for bacteria dysbiosis, there exists fungi dysbiosis in NASH patients. Münevver Demir et al. found that patients with non-obese NASH or F2–F4 fibrosis had distinct fecal mycobiome composition compared to those with mild disease [26]. Antifungal treatment can improve NASH in mice. Thus, intestinal fungi can be an attractive target to attenuate NASH.

The role of gut microbiome dysbiosis on NASH progression can be partially attributed to an increased susceptibility to intestinal permeability. As a result, some pro-inflammatory substances derived from gut microbiota can translocate to the portal vein and liver. For instance, the levels of serum lipopolysaccharide (LPS)-binding protein (LBP) were increased in NASH patients in comparison with NAFLD patients [27]. The increased levels of endotoxin in the portal system and plasma can activate toll-like receptor 4 (TLR4) in the liver of NASH patients. This is further confirmed by the higher level of TLR4+ macrophages in NASH than simple steatosis [28]. TLR4 activation can promote liver macrophage ROS generation and increase expression of pro-interleukin-1β, contributing to a pro-inflammatory environment and finally facilitating NASH process [29]. This indicates the effects of gut microbiota dysbiosis-mediated LPS/TLR4 activation on the pathogenesis of NASH. Aside from this mechanism, a new study has shown that gut-derived microbial antigens can act as ligands that activate the pathogenic function of intrahepatic B cells through MyD88 pathways, giving rise to hepatic inflammation and fibrosis during NASH progression [14].

### 2.3. Gut Microbiome Dysbiosis in NAFLD-HCC

NAFLD-HCC accounts for 10% of all HCC types [30]. NAFLD-HCC arises from chronic inflammation mediated by lipotoxicity, which is due to excessive neutral lipid accumulation in the liver. Considering the effect of gut microbiome on NAFLD progression, it is necessary to elucidate the role of the gut microbiome in NAFLD-HCC. An initial clinical study showed an increased abundance of *Bacteroides* and *Ruminococcaceae* and decreased abundance of *Bifidobacterium* in NAFLD-HCC patients compared to patients with cirrhosis who had not progressed to NAFLD-HCC [31]. In addition, there is an association between the gut microbiota and several inflammatory cytokines such as higher levels of interleukin (IL) 8 and chemokine (C-C motif) ligand (CCL) 3 in NAFLD-HCC patients [31]. This indicates that gut microbiota-driven inflammation may aggravate NAFLD-HCC progression. In addition, the gut microbiota can act as cofactors in the progression of NAFLD-HCC by interacting with immune compartments. A recent study demonstrates that the gut microbiota in 32 NAFLD-HCC patients can reduce the expansion of CD8^+^ T cells but augment the expansion of IL-10^+^ Treg cells compared with 28 NAFLD-cirrhosis and 30 non-NAFLD controls [32]. By establishing a spontaneous NAFLD-HCC mouse model, we have reported that gut microbiota dysbiosis contributes to NAFLD-HCC formation. Dietary cholesterol can drive NAFLD-HCC formation by increasing the abundance of *Mucispirillum, Desulfovibrio, Anaerotruncus* and *Desulfovibrionaceae*, with decreasing levels of *Bifidobacterium* and *Bacteroides* [33].

## 3. Gut Microbial Metabolites in NAFLD and NAFLD-HCC

### 3.1. Gut Metabolomic Changes in NAFLD/NASH

The metabolites of the gut microbiome are the indispensable factor that can modulate the pathogenesis of NAFLD and NASH. Most microbial metabolites are mainly derived from carbohydrate and protein fermentation. Short-chain fatty acids (SCFAs) are one of the most common microbial metabolites derived from indigestible carbohydrates. SCFAs are beneficial to liver metabolism and are involved in NAFLD progression. For example, a recent study found a kind of acetate from a commensal microbe that can suppress NAFLD development by modulating hepatic FFAR2 signaling in the liver of high-fat-fed mice [34]. In addition, several studies demonstrated that another SCFA butyrate was able to attenuate NAFLD by regulating gut microbiota, intestinal tight junctions, hepatic Glucagon-like peptide-1 (GLP-1) receptor expression, and TLR4 pathways [35,36,37,38]. Other SCFA members such as propionate also had a promising capacity of ameliorating NASH progression [39]. However, there is a controversial study showing microbiota-derived acetate can promote hepatic lipogenesis [40]. This may be caused by different diet. In a high fructose diet, the generation of microbial acetate can promote lipogenic pools of acetyl-CoA; however, in a high fat-based diet, acetate can activate FFAR2 signaling to inhibit NAFLD progression.

In addition to SCFA, other types of gut microbial metabolites exhibit a vital role in NAFLD, especially ethanol and bile acid. Microbiome-derived ethanol is from saccharolytic fermentation. Preclinical studies have established that endogenous ethanol derived from microbiota can accelerate liver steatosis and inflammation [41,42]. In addition, clinical evidence has shown that there is an increased level of blood ethanol in NAFLD patients [43]. More importantly, the level of alcohol-producing bacteria in intestine and the concentration of blood ethanol is elevated in NASH patients compared to patients with hepatic steatosis [42]. This indicates that microbiome-derived ethanol is an essential contributing factor promoting simple hepatic steatosis into NASH. Gut microbiome is also involved in bile acid metabolism. The gut microbiota has the capacity of converting primary bile acids into secondary bile acids. In NAFLD, this ability is compromised because of the decreased abundance of related bacteria [44]. A decreased level of deconjugated bile acid can further decrease production of taurine and lead to hepatic steatosis and inflammation by targeting oxidative stress-related genes and fatty acid synthesis-associated genes [45]. In addition, farnesoid X receptor (FXR), the receptor of bile acids, is found out to be downregulated in NAFLD [44]. The decreased level of intestinal FXR can decrease the secretion of fibroblast growth factor 15/19 (FGF15/19), both of which can reduce liver steatosis [46]. Other gut microbial metabolites such as amino acids and choline are also reported to modulate NAFLD [47].

### 3.2. Gut Metabolomic Alteration in NAFLD-HCC

The contribution of gut microbial metabolites to cancer has been well studied in colorectal cancer [48,49]. A few studies have studied the effects of bacterial metabolites on the disease progression in NAFLD-HCC. Recently, a clinical study found that gut microbiota-derived SCFAs such as acetate, butyrate, and propionate, positively correlated with the progression of NAFLD-HCC [32], which is controversial to the findings that SCFAs can alleviate NASH progression. The different effects of SCFAs in NASH and NAFLD-HCC may possibly be attributed to the diverse microenvironments of NASH and NAFLD-HCC. Thus, more investigations are needed to be performed for solving this concern. Except for clinical evidence, a recent study from our team showed that there is an alteration of gut bacterial metabolites such as increased level of serum taurocholic acid (TCA) and decreased level of serum 3-indolepropionic acid (IPA) in mice with NAFLD-HCC [50]. Mechanistically, we found that IPA was able to inhibit lipid accumulation and cell proliferation, while TCA promoted triglyceride accumulation in the liver. So far, few studies have investigated the correlation between NAFLD-HCC and gut metabolomic changes, more research is needed to elucidate the possible role of gut microbial derived metabolites in the NAFLD-HCC progression.

## 4. The Communication between Gut and Liver in NAFLD and NAFLD-HCC

### 4.1. Intestinal Permeability

Intestinal permeability determines what can be transported from gut to liver and influences NAFLD progression. Intestinal permeability is dependent on the intestinal barrier consisting of mucus layer, intestinal epithelium, mucosal immune system, and the gut vascular barrier (GVB). The central roles of the intestinal barrier are enterocytes and GVB in charge of entry into the portal vein and access to the liver.

Enterocytes are tightly connected to each other by junctional proteins including E-cadherins, occludins, claudins, and junctional adhesion molecules [30]. Gut microbiota can reinforce intestinal integrity by producing metabolites such as SCFA, which can directly reinforce tight junctions, while gut microbiota dysbiosis can lead to compromised gut barrier integrity. In NAFLD patients, the existed gut microbiota dysbiosis can easily disrupt tight junctions and cause increased intestinal permeability [51,52,53]. In contrast, some bacteria may prevent NAFLD by influencing intestinal epithelium connection. For instance, increased abundance of *Akkermansia muciniphila* is related to improved gut permeability and NAFLD progression by regulating tight junctions [54,55].

The disruption of intestinal tight junctions can cause translocation of bacteria and their metabolites from gut lumen to lamina propria. In the lamina propria, an intact GVB, which consists of blood endothelial cells, can prevent bacteria or its toxic metabolites from reaching the portal circulation. Whereas, during high-fat diet-induced dysbiosis, GVB is compromised at a very early stage. The impaired GVB can promote bacterial translocation to the liver and promote hepatic steatosis, inflammation, and fibrosis [56]. More importantly, improving GVB by FXR agonist obeticholic acid (OCA) can ameliorate NASH in mouse model and patients [26,57]. OCA can improve NASH by enhancing GVB function and reducing bacterial translocation in a mouse model [57]. A multicenter, randomized, placebo-controlled phase 3 clinical trial including 1968 NASH patients showed that OCA could improve NASH activity and fibrosis for NASH patients [58]. This indicates targeting GVB by OCA is a promising approach for NASH treatment. In conclusion, breakdown of these gut barrier components by diet and inflammation can increase intestinal permeability. Thus, microorganisms and microorganism-derived molecules can easily enter the portal vein and finally reach to liver. This event can be considered as the first step of communication between gut and liver in NAFLD.

### 4.2. Portal Vein Circulation

The portal vein is a vital circulation system that directly connects the liver with the intestine. Approximately 70% of the blood supply in the liver is derived from the portal vein, which drains blood from intestinal mesenteric veins [59]. Under normal conditions, many nutrients and beneficial microbial products can reach the liver via the portal system. For example, microbiota-derived SCFAs can be absorbed and enter into the liver through the portal system [60]. In some pathological conditions such as gut inflammation and dysbiosis, the disrupted gut barrier leads to gut-derived toxic factors translocated into the portal tract, including live bacteria, damaged bacterial components (damage-associated molecular patterns (DAMPs), LPS), and proinflammatory bacterial metabolites (ammonia and ethanol). Through the portal system, these toxic factors can directly enter the liver, stimulate immune cells and the inflammatory cytokine pathways, and finally lead to NAFLD development [61]. A preclinical study found that high-fat diet fed mice with intestinal inflammation can elevate microbial-derived LPS levels in the portal system and promote NASH progression [62]. Clinically, NASH patients can detect increased LPS level in the portal tract compared with those with simple steatosis [28]. Except for NASH, there is metabolic alterations in HCC patients in the portal system as well. A recent study from our group demonstrated that higher level of DL-3-phenyllactic acid, L-tryptophan, and glycocholic acid can be detected in HCC patients in the portal vein in comparison with healthy controls [63]. These studies elucidate the central role of portal vein circulation on the communication between gut and liver in NAFLD.

### 4.3. Bile Acid Circulation

Bile acid circulation is another essential enterohepatic circulation in NAFLD. Bile acids (BAs) are steroid molecules that are synthesized from cholesterol in hepatocytes by approximately 15 enzymes. The BAs secreted from the liver cells can flow through the biliary tree and into the gallbladder, where the BAs can be released into the intestine during the inter-digestive period [64]. In the intestine, gut microbiota can transform and esterify BAs by utilizing bile salt hydrolases (BSH). BSH is active in a variety of bacterial genera such as Lactobacillus, Bifidobacterium, Clostridium, and Bacteroides, which can produce free BAs. Free BAs can solubilize intestinal lipids and are less efficiently reabsorbed. Thus, a high level of BSH is closely related to the reduction in body weight, and a reduced serum cholesterol level and liver triglycerides level [65]. In NAFLD, the interaction between gut microbiota and BA metabolism critically impact the NAFLD progression. For instance, NASH patients with gut dysbiosis increased BA synthesis [66]. A direct evidence was provided by the antibiotic treatment. A preclinical study showed that antibiotic treatment can regulate the bile acid/intestinal FXR axis and lead to increased hepatic lipids [67]. In addition, germ-free mice can also provide direct evidence about the relationship of gut microbiota and BA circulation in NAFLD. Germ-free mice are found resistant to HFD-induced obesity, while gut microbiota can increase weight gain and liver lipid accumulation of the mice by FXR-dependent mechanisms [68]. Hence, targeting intestinal FXR can be an effective therapeutic target for NAFLD treatment. However, since most of these studies are on the basis of animal studies, it is still unclear whether there are differences between mouse and human in affecting microbial BA metabolism. Therefore, more human studies should be conducted on the complicated communications among gut microbiota, BAs, and NAFLD pathogenesis.

### 4.4. Intestinal Hormones

Apart from direct communication between the gut and liver by portal system and bile acid circulation, gut microbiota can also contribute to the liver metabolism through influencing the secretion of intestinal hormones with the ability to enhance glucose-induced insulin and to inhibit glucagon release. For example, the microbiome derived SCFA can trigger the secretion of glucagon-like peptide-1 (GLP-1), which is an intestinal hormone secreted by the intestinal L cells [69]. Many GLP-1 receptor agonists have shown the capacity of reversing hepatic steatosis and serving as a new alternative for NAFLD treatment [70,71]. Moreover, gut microbiota analysis found that liraglutide, a GLP-1 receptor agonist, could modify gut microbiota diversity by decreasing the abundance of Proteobacteria and increasing the level of *Akkermansia muciniphila*, which was associated with the improvement of NAFLD [72]. In addition to GLP-1, other intestine hormones such as fibroblast growth factor 15 and 19 (FGF15 and FGF 19) can also ameliorate HFD-induced hepatic steatosis with the engagement of gut microbiota [46,73]. In addition, gut microbiota can influence another L cell-derived gut hormone called insulin-like peptide 5 (INSL5), which is found to be involved in pathophysiology of NAFLD [74]. On the basis of these studies, NAFLD development could be impacted by the interaction between gut microbiota and intestinal hormones in the gut–liver axis. However, there is still a lack of clinical evidences to support the communication between gut microbiota and gut hormones in NAFLD.

## 5. Prevention and Therapeutic Strategies of NAFLD by Modulating Gut Microbiome

### 5.1. Probiotics

Modulation of gut microbiota by probiotics is an emerging and promising therapeutic method for the malfunction of the gut–liver axis and NAFLD. The FAO/WHO defines probiotics as live microorganisms with health benefits on the hosts, when being administered in adequate amounts [75]. In the aspect of the gut–liver axis, the protective effect of probiotics on NAFLD acts mainly by improving the gut barrier. For example, several probiotics such as *Lactobacillus rhamnosus GG, Lactobacillus acidophilus*, *Lactobacillus plantarum*, and *Streptococcus thermophilus* have shown the capacity of activation of tight junction proteins to improve the intestinal permeability [76,77]. More recently, a study demonstrated that probiotics are capable of stabilizing the mucosal immune function, which protect NAFLD patients from increased intestinal permeability [78].

Clinically, a variety of probiotics has been investigated on the prevention and treatment of NAFLD, especially common probiotics such as *Lactobacillus*, *Bifidobacterium,* and *Streptococci*. Wong et al. conducted a 6-month treatment with a mixture of different probiotics for NASH patients and found that there is the significantly reduced fat content in the liver compared to the placebo group in subjects treated with probiotic [79] (Table 1). Clinical evidence also indicates that probiotic can not only improve liver histology, but also alleviate liver injury index such as aspartate aminotransferase (AST) and alanine aminotransferase (ALT) in NAFLD patients [80]. However, in the same year, another clinical trial found that treating NAFLD patients with multiple-strain probiotics can improve only liver steatosis but not liver enzymes [81] (Table 1). These studies indicate that more elaborate probiotic pharmacotherapies need to be provided regarding to the efficacy and safety profiles of probiotics in clinical practice.

### 5.2. Prebiotics

Prebiotics are defined as the food components that have beneficial effects on the host associated with the modulation of the microbiota [82]. Common prebiotics include fructooligosaccharides (FOS), inulin, transgalactooligosaccharides (TOS), and lactulose. Prebiotics are able to increase the growth and activity of probiotics, thereby being an effective and safe method of regulating the gut microbiota [83]. For instance, prebiotics can inhibit the growing of pathogenic bacteria such as Salmonella enteritidis, Klebsiella pneumoniae, as well as Escherichia coli; and meanwhile, it can activate the beneficial bacteria [84]. This property can further promote gut microbiota homeostasis, improve the gut barrier, and finally ameliorate NAFLD progression. Prebiotic can also serve a protective role in NAFLD through fermentation to produce SCFAs, including acetate, propionate, and butyrate, which have been well studied in protecting the gut–liver axis and NAFLD [83]. More recently, Sun et al. found a new soluble dietary fiber called Larch wood arabinogalactan (LA-AG) as a candidate prebiotic. LA-AG was capable of increasing the accumulation of organic acids by fermentation to inhibit the activity of pathogenic bacteria and improve gut health [85]. Thus, LA-AG may have the potent to the prevention of NAFLD through regulating gut–liver axis.

In the clinical trials, many prebiotics have been found to present beneficial effects on NAFLD patients. Considering oligofructose as an example, Bomhof et al. demonstrated that the supplement of the prebiotic oligofructose can improve liver steatosis and non-alcoholic fatty liver activity score (NAS) in patients with NASH [86]. In addition, a recent meta-analysis summarized that prebiotic supplementation for NAFLD patients can improve their anthropometric and biochemical parameters, including body mass index (BMI), ALT, AST, fasting insulin, as well as insulin resistance [87] (Table 1). Nevertheless, the duration of future clinical studies should be for a sufficient time so as to better elucidate the potential of prebiotics as a future treatment for NAFLD and NASH patients.

### 5.3. Antibiotics

The application of antibiotics in the NAFLD treatment is based on the concept that antibiotics can diminish the influences of microbiota and their metabolites on the host metabolism through the gut–liver axis. In 2008, there was a study showing that neomycin and polymyxin B can markedly reduce hepatic lipid accumulation by reducing the translocation of endotoxin in an NAFLD mouse model [88]. Additionally, another preclinical study found administration of antibiotics can regulate the level of portal secondary bile acid by suppressing the gut bacteria, thereby attenuating inflammation and fibrosis in the liver, and thus protecting NAFLD progression [89]. Clinically, antibiotics also show a promising efficiency in preventing NAFLD. For instance, Solithromycin, a potent next-generation macrolide antibiotic, was found to reduce ALT and NAS of NASH patients in a Phase II clinical trial [90] (Table 1). However, antibiotics should be cautiously used since they could eliminate some important bacterial species related to healthy status and lead to the presence of some antibiotic-resistant bacteria [91].

**Table 1 biomedicines-10-00524-t001:** Microbiota-based treatment targeting gut–liver barrier for NAFLD.

Treatment Methods	Beneficial Effects	Clinical Study
Probiotics: *Lactobacillus plantarum, Lactobacillus deslbrueckii, Lactobacillus acidophilus, Lactobacillus rhamnosus,* and *Bifidobacterium bifidum,* etc.	Increasing intestinal barrier integrity, stabilizing the mucosal immune function	Liver steatosis [81], NASH [79]
Prebiotics: oligofructose, inulin, *Ocimum basilicum*, psyllium	Promoting gut microbiota homeostasis, promoting fermentation of beneficial metabolites, Increasing intestinal barrier integrity	Liver steatosis [87], NASH [86]
Antibiotics	Diminishing the pathogenic microbiota and metabolites (endotoxin)	Liver steatosis (NCT02009592), NASH [90].
FMT	Re-establishing a balanced gut microbiota, increasing intestinal barrier integrity	Liver steatosis (NCT02009592), NASH (NCT02469272)

### 5.4. FMT

FMT, a therapeutic approach transferring fecal microbiota and metabolites from healthy donors to patients who need to re-establish a balanced gut microbiota, is an emerging treatment method for gastrointestinal disease, such as Clostridium difficile infection [92]. Several studies have shown that FMT is also an efficient bacteriotherapy for NAFLD. In an early preclinical study, Zhou et al. found that FMT could alleviate HFD-induced NASH by regulating gut microbiota, increasing SCFA levels, and improving the gut barrier [93]. More recently, our group showed there is a lower hepatic lipid accumulation and inflammation in germ-free (GF) mice receiving FMT from normal chow-fed mice compared to those receiving FMT from high-fat/high-cholesterol (HFHC) fed mice [50]. Consistent with animal experiments, recent clinical trials also found that FMT can reduce hepatic steatosis and intestinal permeability in NAFLD patients [94]. However, there are still some adverse events reported in FMT such as bacteremia and perforations [95,96]. Thus, more clinical trials should be conducted to improve the efficacy and reduce the side effects of FMT treatment in NAFLD/NASH.

### 5.5. Gut Microbiome-Based Personalized Therapy

The gut microbiome is a critical component in personalized medicine. Gut microbiome-based personalized therapy can provide personalized therapeutic interventions in NAFLD by modulating personalized microbiome changes. It includes targeting gut barrier integrity, targeting intestinal dysbiosis, targeting gut microbial metabolism, and targeting personalized nutrition [97]. The selection of different gut microbiome-based therapeutic methods for NAFLD patients is on the basis of microbiome-based stratification dependent on the microbial feature of patients’ metagenome data and metabolite data [98]. In addition, patient-level factors should also be considered in the personalized therapy such as patient age, sex, and clinicopathological parameters. Nevertheless, there is still insufficient studies investigating personized microbiome-based approaches to NAFLD management and prevention.

## 6. Conclusions and Future Perspective

The gut–liver axis provides an important bridge between gut and liver. The dysbiosis of gut microbiota can alter intestinal permeability, increase the level of portal toxic metabolites, promote hepatic inflammation, and thus lead to NAFLD and NAFLD-HCC development. A variety of studies suggest that microbiota-based pharmacological modulation targeting the gut–liver axis is a promising and helpful therapeutic method for NAFLD treatment. The gut–liver axis plays a role in the gut microbiota dysbiosis and microbiome-based treatment of NAFLD (Figure 2). However, the microbiota-based treatment is still in the preclinical stage for NAFLD-HCC patients, and thereby deserving of more clinical investigations. In addition, based on the concept of personalized microbiota treatment, future research needs to pay more attention to the development of specific probiotics, beneficial bacterial metabolites, or inhibitors targeting specific pathogenic microbes and metabolites for NAFLD and NAFLD-HCC.

## Figures and Tables

**Figure 1 biomedicines-10-00524-f001:**
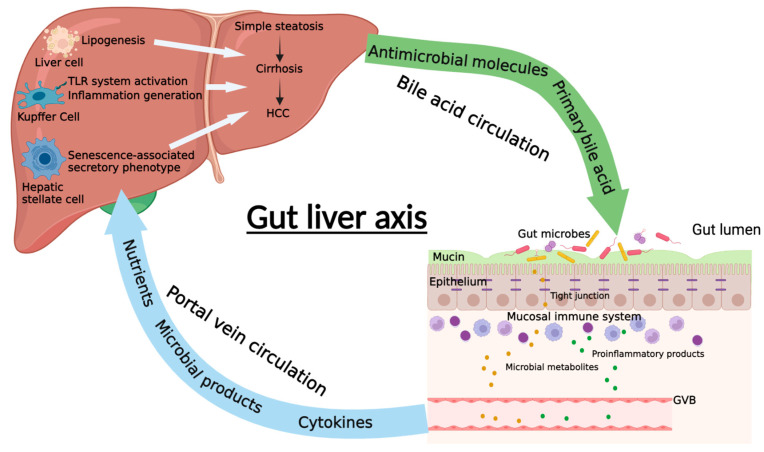
The diagrammatic representation of gut–liver axis. The communication between gut and liver in NAFLD is the intestinal barrier, which is composed of mucus layer, intestinal epithelium, mucosal immune system, and gut vascular barrier (GVB). The gut microbiota and their metabolites as well as some proinflammatory products can pass through the intestinal barrier and enter the liver through the portal system. In the liver, they can promote or inhibit the progression of NAFLD through different mechanisms. Meanwhile, liver can also regulate intestinal function and gut microbiota balance through the bile acid circulation, which is an important enterohepatic circulation in regulating NAFLD (Created with BioRender.com).

**Figure 2 biomedicines-10-00524-f002:**
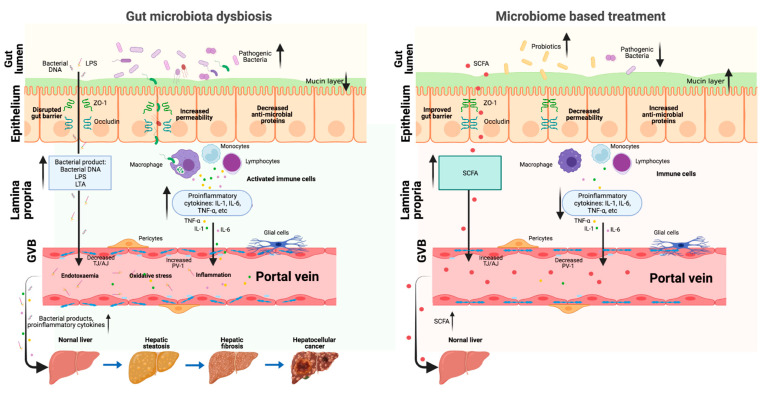
The role of gut–liver axis in the gut microbiota dysbiosis and microbiome-based NAFLD treatment. When the gut microbiota is in homeostasis, it is difficult for the pathogenic bacteria and metabolites to go through the intestinal barrier and reach the liver. When the gut microbiota is dysbiosis, the intestinal barrier is damaged (decreased mucus layer, increased epithelium, and endothelium layer permeability); thus, the pathogenic bacteria and metabolites can easily go through the intestinal barrier, come to the portal system, and finally reach the liver, where they can promote NAFLD progression. Microbiome-based treatment (probiotics and beneficial metabolites) can repair the intestinal barrier and the beneficial metabolites can reach the liver through the portal vein to prevent NAFLD progression (Created with BioRender.com, https://biorender.com/, accessed on 4 February 2022).

## Data Availability

Not applicable.

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
