# Peer review of "The Role of Gut–Liver Axis in Gut Microbiome Dysbiosis Associated NAFLD and NAFLD-HCC"

_biomedicines, 2022, doi:10.3390/biomedicines10030524_

Round 1

Reviewer 1 Report

The manuscript is well written, with a correct sequence in the information illustration and the level is appropriate to readership. The topic is very current. Recent studies show that microbiota dysregulation plays a key role in the pathogenesis of NAFLD and NASH, via the gut-liver axis. However, as already reported in my revision, the review is not as original as a very similar review, on the same topic and with a similar outline, was published very recently (Int.J. Mol. Sci. 2021). Perhaps the authors could have further developed the role of FXR agonist (OCA) that has been shown to ameliorate NASH in animal models and in patients. The interest of this paper is the therapeutic strategies, gut microbiome modulated, of NAFLD. The section is well structured and well summarizes the most recent strategies.

Author Response

Response to Reviewer 1 Comments

Comments are written in italics and responses in plain texts.

The manuscript is well written, with a correct sequence in the information illustration and the level is appropriate to readership. The topic is very current. Recent studies show that microbiota dysregulation plays a key role in the pathogenesis of NAFLD and NASH, via the gut-liver axis. However, as already reported in my revision, the review is not as original as a very similar review, on the same topic and with a similar outline, was published very recently (Int.J. Mol. Sci. 2021). Perhaps the authors could have further developed the role of FXR agonist (OCA) that has been shown to ameliorate NASH in animal models and in patients. The interest of this paper is the therapeutic strategies, gut microbiome modulated, of NAFLD. The section is well structured and well summarizes the most recent strategies.

Response: We thank the reviewer for the positive comments and very useful suggestion. The detail discussion on the role of FXR agonist (OCA) in ameliorating NASH by improving gut barrier function has now been added on Line 249-253.

Reviewer 2 Report

This paper is a comprehensive, interesting, and well-structured review of the role of the gut-liver axis in gut microbiome dysbiosis associated with NAFLD and NAFLD-HCC.

Minor suggestions,

As the authors discuss in their paper, also about gut dysbiosis in NAFLD up to HCC, I suggest including into your title this aspect.

Recently, experts reached a consensus that NAFLD does not reflect current knowledge, and metabolic (dysfunction) associated fatty liver disease "MAFLD" was suggested as a more appropriate overarching term. I suggest including a small phrase/sentence in your introduction about this.

Row 41 Rephrase this sentence as the use of “human cells” seems repetitive.” The gut microbiota has been considered as a critical partner of human cells, which can interact with a variety of human cells”

Row 46, should be a reference after this sentence? :“Since then, emerging evidence has shown the critical effects of gut microbiota on the maintenance of host metabolism”.[?]

Row 151, a reference should be added after this sentence “there is an association between the gut microbiota and several inflammatory cytokines like..”

Avoid the use of this phrase: “In summary”, in sentences near of one another. In row 417 and at the start of conclusions.

Generally, in conclusions, I suggest avoiding phrases such as: “It is clearly illustrated in Figure 2”

Figures and Tables should be placed into the main text close to their first citation.

Use Justify function for your text.

Author Response

Response to Reviewer 2 Comments

This paper is a comprehensive, interesting, and well-structured review of the role of the gut-liver axis in gut microbiome dysbiosis associated with NAFLD and NAFLD-HCC.

Minor suggestions,

1. As the authors discuss in their paper, also about gut dysbiosis in NAFLD up to HCC, I suggest including into your title this aspect.

Response: We appreciate the comment. We have now revised the title to “The role of gut-liver axis in gut microbiome dysbiosis associated NAFLD and NAFLD-HCC”.

2. Recently, experts reached a consensus that NAFLD does not reflect current knowledge, and metabolic (dysfunction) associated fatty liver disease "MAFLD" was suggested as a more appropriate overarching term. I suggest including a small phrase/sentence in your introduction about this.

Response: Thanks for your suggestion. We have now added the Introduction of MAFLD in the revised manuscript on Line 35-41.

3. Row 41 Rephrase this sentence as the use of “human cells” seems repetitive.” The gut microbiota has been considered as a critical partner of human cells, which can interact with a variety of human cells”

Response: We appreciate the comment. We have now rephrased the sentence to “Gut microbiota has been considered as an indispensable metabolic organ, which can interact with host cells” (now Line 46-47).

4. Row 46, should be a reference after this sentence? :“Since then, emerging evidence has shown the critical effects of gut microbiota on the maintenance of host metabolism”.[?]

Response: We have now added the reference for this sentence (now Line 52).

5. Row 151, a reference should be added after this sentence “there is an association between the gut microbiota and several inflammatory cytokines like..”

Response: We have now added the reference for this sentence (now Line 170).

6. Avoid the use of this phrase: “In summary”, in sentences near of one another. In row 417 and at the start of conclusions.

Response: Thanks for the comment. We have removed “In summary” in the revised text.

7. Generally, in conclusions, I suggest avoiding phrases such as: “It is clearly illustrated in Figure 2” Figures and Tables should be placed into the main text close to their first citation.

Response: We have now placed the tables and figures in brackets in which they are first mentioned in text.

8. Use Justify function for your text.

Response: Thanks for the suggestion. We have justified all text.